# Characterization of Adolescent Pregnancy and Legal Abortion in Situations Involving Incest or Sexual Violence by an Unknown Aggressor

**DOI:** 10.3390/medicina55080474

**Published:** 2019-08-13

**Authors:** Maria Misrelma Moura Bessa, Jefferson Drezett, Fernando Adami, Sandra Dircinha Teixeira de Araújo, Italla Maria Pinheiro Bezerra, Luiz Carlos de Abreu

**Affiliations:** 1Laboratório de Delineamento de Estudos e Escrita Científica, Centro Universitário Saúde ABC, Santo André 09060-870, Brazil; 2Programa de Mestrado em Políticas Públicas e Desenvolvimento Local da Escola Superior de Ciências da Santa Casa de Misericórdia, Vitória 29045-402, Brazil; 3Programa de Mestrado em Ciências da Saúde da Amazônia da Universidade Federal do Acre. Bolsista Capes Brasil, Rio Branco 69.915-900, Brazil; 4Graduate Entry Medical School, University of Limerick, V94 T9PX Limerick, Ireland

**Keywords:** incest, sexual violence, pregnancy in adolescence, induced abortion

## Abstract

*Background and Objectives:* In pregnancies resulting from incest, the adolescent maintains close family and emotional relations with the aggressor, different from what occurs when pregnancy results from sexual violence by strangers. Evidence indicates that this type of relationship with the aggressor may interfere in the dynamics of such violence and the adolescent’s access to health services. *Materials and Methods:* The objective of this research was to describe and correlate aspects associated with pregnancy when resulting from rape of adolescents in situations of incest; rape when perpetrated by an unknown aggressor and an abortion as allowed by law was sought. Method: A cross-sectional, epidemiological study of adolescents treated at the Pérola Byington Hospital, São Paulo, Brazil, bringing an allegation of pregnancy, resulting from sexual violence and a request for abortion as allowed by law. A total of 311 adolescents, being 134 in the “pregnancy from incest group”, and 174 in the group “pregnancies resulting from rape by a stranger” were considered under the study variables; relationships were investigated using the chi-squared test and Poisson regression with robust variance. *Results:* The study included 137 cases (44.1%) of pregnancy resulting from incest, and 174 cases (55.9%) of pregnancy from rape by a stranger. In cases of incest, a declaration of religion (92.0%) was significantly more frequent, and the adolescents were approached in spaces considered safe or private (92.7%); the aggressor taking advantage of the adolescent’s legal condition of vulnerability as a function of age (83.3%). Cases of incest presented a lower median adolescent age and greater gestational development, with gestations being ≥ 13 weeks prevailing. *Conclusion:* Cases of pregnancy by incest presented indicators suggesting both proximity and relationship with the aggressor, and pregnancy at a very early age, which postponed the adolescent’s procurement of health service, and interfered negatively with abortion assistance as allowed by law.

## 1. Introduction

In the history of societies, incest has suffered conceptual variations in function of the historical moment, culture, social class, and religion. It expresses a particular form of power relationship [1].

Psychological, social, and psychiatric theories, upon involving with incest as a concept have fostered amplifications concerning its consequences for evolutionary processes in families and societies [2]. Incest is considered violation of a taboo prohibiting sexual relations between members of the nuclear family, (excluding husband and wife), and extending to others of differing degrees of kinship, whose ties may be either biological or simple affinity [2].

Current understandings of incest must overcome the reductionism implicit in legal definitions; these often limit the phenomenon to biological or kinship degrees [3]. To a certain extent, incest is an expression of gender-related violence, since it is based on power asymmetry; between the aggressor and the victim.

Sexual violence predominantly affects female adolescents, although results differ concerning consequences between genders [4]. In cases of incest, the phenomenon often repeats, possibly resulting from patriarchal cultures that constitute the most ancient social formations [5].

Although sexual violence can affect women at any stage of life, cases of incest concentrate during childhood and adolescence. The consequences in this age group may be even more severe, since the violence affects these persons in transitional phases involving prominent psychosocial changes [4]. Though incest can produce many health injuries, victims of reproductive age can often face the problem of pregnancy. In these cases, the adolescent maintains close relations with the aggressor, different from what occurs in pregnancies where a stranger is responsible for the sexual violence [6].

Incestual relations can therefore interfere in the outcome of these pregnancies because of the aggressor’s role. According to Blake et al. [7], the delayed search for health services is more common when pregnancy results from sexual violence practiced by known or family aggressors. Thus, it would be reasonable to assume that in pregnancies resulting from incest, the proximity of the aggressor might hinder or delay revealing both the violence and the pregnancy, thus affecting the adolescent’s success in obtaining assistance.

In situations of incest, the close relationship with the aggressor may well be a factor that inhibits the communication of sexual violence to the police, particularly for the need to maintain the “family secret” and avoid the perpetrator accountability. In Brazil, such resistance can affect submission to a legal medical examination, since the procedure is required exclusively by the police and depends, therefore, on the police communication officer [7].

There is also evidence that the victim/aggressor relationship can influence the mechanisms used to impose violence on such adolescent girls. While sexual crimes committed by strangers in circumstances of urban violence may be associated with an increased frequency of physical violence, adolescents in an incestual situation may be more prone to acts of coercion, threats, psychological abuse, fear, or even affected by respect for the privileged position of the aggressor within the family.

There is little knowledge concerning the social-demographic characteristics of pregnant adolescents suffering incestuous sexual violence. Further, it is not sufficiently clear whether proximity to the aggressor interferes in legal abortion assistance when required by the victim. The objective of this research was to describe and correlate aspects associated with pregnancy when resulting from rape of adolescents in situations of incest, and rape when perpetrated by unknown aggressors, and a request for abortion as allowed by law was sought.

## 2. Methods

A Cross-sectional epidemiological study was conducted using a database consisting patient information of those treated between July 1994 and December 2015 at Pérola Byington Hospital, Public Health Service of the State of São Paulo. This hospital, the legal reference hospital for situations of sexual violence and abortion, considers as adolescents, girls aged 12 through 17 years, as defined by the Child and Adolescent Bylaw [8].

These two articles characterize situations of sexual violence that allow abortion in Brazil, as laid down in Penal Legislation Article 128. Article 213 defines as rape any non-consensual sexual act committed by the use of physical force or serious threat. Article 217-A, which deals with rape of vulnerable persons, covers sexual acts against persons under the age of 14 or against those of any age who cannot offer resistance or valid consent [9].

For the study, the adolescents were allocated into two groups: (A) Adolescents with pregnancy resulting from incest, and (B) Adolescents with pregnancy resulting from sexual violence perpetrated by strangers. The incest concept adopted was the manifestation of sexual relationships between people who are members of the same family (except spouses), and the family condition was defined by consanguinity, affinity, and social function of kinship exerted by a person within the group [10].

Thus, pregnancies resulting from incest were those in which the aggressor was the father, stepfather, brother, uncle, grandfather, cousin or brother. The study excluded cases of sexual violence perpetrated by known aggressors, which were not classifiable as incest.

The data were extracted from a database organized in Microsoft Excel 2010 and transferred to Stata 11.1. The database was completed using a pre-coded supplementary sheet. Differing referees reviewed the consistency of the information included in each case. The data were examined; and discrepancies identified and corrected prior to the transfer to Stata 11.1.

Incest was adopted as the dependent variable against cases involving an unknown aggressor. The study variables were analyzed according to these two categories. The socio-demographic characterization of adolescents was performed using the following variables: age group, ethnicity; religious belief, and form referral to the hospital.

The characterization of the sexual crime was analyzed concerning intimidation, legal condition of vulnerability, event bulletin, forensic examination by an Institute of Forensic Medicine, and location where the adolescent was approached. The characteristics of the pregnancy and the outcome of the abortion request were evaluated according to gestational age; reasons given to not perform or not approve the abortion; and the method adopted for interrupting the pregnancy.

For statistical analysis, frequencies and percentages were calculated to describe the adolescents’ profile and to estimate the prevalence of incest cases. The adolescents’ age and the gestational age were described using mean and median, respectively, according to data normality assessed by the Shapiro-Wilk test.

Correlations between cases of incest and profile factors associating the adolescents and the environments where the rapes had occurred were investigated by prevalence ratio at 95% confidence intervals using the Chi-square test and Poisson regression with robust variance. Associations between the cases of incest for the adolescents’ age and gestational age were assessed (respectively) using the Student t test and the Mann-Whitney test, according to the normality of the observed data.

The choice of specific tests was performed according to data normality and by comparison of interval estimates [11]. In the cross-sectional studies, the Poisson regression adjusted using a robust variance model presented better consistency for the estimates as compared to the logistic regression, and for these motives was chosen [12].

The Research Ethics Committee of Juazeiro do Norte College, (protocol 1.287.857), approved the study. To ensure confidentiality, the study did not include registration of the adolescents’ identities. All adolescents received medical, social, and psychological advice, including guidance on legal and ethical issues relating to abortion and sexual violence. The Research Ethics Committee of Juazeiro do Norte College approved this study; however, the authors are not affiliated with this institution.

## 3. Results

The study included 311 adolescents with allegations of pregnancy resulting from sexual violence between the 1994 and 2015, being 137 cases (44.1%) of incest and 174 cases (55.9%) of sexual violence perpetrated by strangers. Table 1 describes the socio-demographic characteristics and sexual violence (Table 1).

Table 2 presents age analysis for both groups, whose data presented normal distribution. The mean age was 1.2 years lower in adolescents with allegations of pregnancy by incest than for those with allegations of pregnancy by strangers. The gestational age did not have a normal distribution; in the incest group, it was greater than in the group who had suffered violence by strangers (*p* < 0.001).

Table 3 presents the data on the outcome of the pregnancy in both groups studied. For gestational age, there was predominance in pregnancies of over 12 weeks in the incest group as compared to the gestational age of the pregnancy by stranger category.

Poisson regression (Table 4) indicates the association of variables which refute or confirm the hypotheses of the study. Other variables deserve special mention, such as the form of intimidation prevalent in the incest group, with rape of the vulnerable being the more frequent. Among the aspects of rape of vulnerables, the predominant prevalence ratio an age of less than 14 years had greater relevance.

Regarding gestation characteristics, gestational age presented a delay in search for care; the prevailing gestational age was above 23 weeks, which affected the variable of gestation non-approval because of advanced gestational age, making legal abortion impossible.

## 4. Discussion

The perspective of pregnancy outcomes due to incest allows association of several aspects. Variants such as religious belief, police participation, adolescent age, and gestational age when seeking care may contribute to clarify important characteristics of the event when comparing these variables with those of adolescents suffering from sexual violence by strangers. The results of this research point to consistent differences in these two groups, especially in gestational outcomes.

The differences refer to the age range of seeking the healthcare service to perform an abortion; when it is impossible to perform it, since in Brazil the legislation limits abortion to 22 weeks. In relation to victims of an unknown aggressor, the more advanced gestational age occurs in incest victims.

Another variable of great importance is the type of intimidation used against the groups. The characterization of rape of the vulnerable under Brazilian law is more prevalent in the incest victim group, pointing to the condition of lesser self-defense ability when the aggressor is a relative, since he enjoys the intimacy of relationship with the victim.

This study began with the fundamental element of sexual violence, occurring in opposing contexts. At one extreme, incest is practiced within the family, where the aggressor takes advantage of trust and intimacy with the adolescent to initiate and maintain the sexual abuse. At the other extreme, the sexual violence practiced by a stranger, (without any interpersonal relationship with the adolescent), configures as a violent crime common in large urban centers [13].

In situations of alleged pregnancy resulting from violence, declaration for religion by the adolescent was highly frequent in both groups, although it was significantly higher in incest cases. However, a simple statement of religion or any religious practice, does not allow analyzing its influence as a factor relating to incest. However, the higher frequency of declaration of religion among the incest cases removes the popular belief that incestual sexual violence occurs more often in agnostic families. An American study conducted by Gervais (2014) lists incest as an immoral act, considered so, regardless of religious belief, including by those who reported being atheists [14].

For the criterion of ethnicity or skin color of the adolescent, there was no difference between the groups, not allowing analysis from an ethnic-racial viewpoint. This finding has important implications in a psychological context and for its more traditional authors, where the incest taboo is considered to be weaker among non-whites [15].

Certain study variables indicated that the relationship between the aggressor and the adolescent produces clear effects on the total process of the pregnancy. The adolescents’ ages in incest pregnancies were on average, 1.2 years lower than those observed for those who suffered sexual violence perpetrated by strangers. In this case, the dynamics of incest must be considered, usually characterized by a recurrent sexual conduct often initiated during childhood [16].

When including acts of vaginal penetration, incest exposes the adolescent repeatedly to the grievance of pregnancy, and may explain the higher precocity of pregnancies found in this study. In addition, the family, community, and public authorities often only perceive incest if a pregnancy occurs; the clinical condition cannot be hidden and reveals the family secret. In fact, there is evidence that the difficulty identifying and recognizing incestuous sexual abuse is directly proportional to the degree of kinship between the victim and the aggressor [17,18].

Authors, such as Zambon et al. [19], found that, in cases of chronic sexual violence, the chance of an incestuous aggressor was nearly four times greater. A study conducted by Yildirim et al. (2014) corroborates this chronicity of sexual violence in such relationships, showing that one third of incest cases occur for more than a year [16].

The data found in this study corroborate the consensus that incest is predominantly intra-familiar, and the chronicity of these incestuous situations presents a clear relationship to the type of approximations that occur.

Within the private space, the aggressor articulates the means necessary to secretly establish the sexual violence, and to protect himself from accountability [18].

On the other hand, in sexual violence as imposed by strangers, the adolescent is generally approached during daily activities performed in public spaces. Zambon et al. (2012) showed that 72.6% of sexual abuse cases against children and adolescents occur in a domestic environment, being 80.1% by a known aggressor and 31.6% being incestuous [19].

Intrafamily violence, or violence within the family, involves blood relatives who may (or may not) occupy the same residence; although the probability of occurrence of incest is greater among relatives living daily in the same household. Domestic violence, in turn, is not limited to the family and involves all the people who live in the same domestic space, bound or not by ties of kinship [20].

To demystify certain ideas spread in a conservative society. Neither blood bonding nor sexual orientation dictates the occurrence or frequency of sexual abuse. Traditional culture in the education of children imposes and dictates warnings or carefulness concerning strangers. However, it is not common to warn children about protecting their bodies from their most intimate living partners, including parents or relatives.

Pregnancy due to incest, by all of the factors listed, does not always decisively end the cycle of violence. Since it is a sort of taboo, it can remain a family secret; the difficulty of identifying the incest may be common to all concerned, even in situations that result in gestation.

Incestuous sexual abuse is the more difficult complaint to be revealed and faced. When it occurs outside family relations, the complaint of sexual violence seems to be more frequent and the credibility of the victim’s narrative is greater. This does not occur when the abuser is kin. The family often makes an effort to maintain the alleged family harmony and normality, which would be threatened by the revelation [17]. The domestic environment, thus ceases to be a protective environment for the adolescent and the aggressor begins to benefit from the social taboo of incest, making the sexual crime invisible to the community, and to civil protective organizations.

Nevertheless, the grievance of pregnancy resulting from sexual violence seems to modify the secret scenario [21]. In both studied groups, in most of the cases, communication to the police by means of event bulletin was similar and complete; suggesting that pregnancy by incest somehow motivates the victim and her family to denounce the aggressor, in a way similar to sexual crimes by strangers.

In both groups, a high percentage of forensic examination completions by the Forensic Medicine Institute were observed. In Brazil and in many countries, this element brings evidence, and is part of the procedures considered necessary to bring accountability to the aggressor, and it suggests the perception by the victim and her family that sexual violence is a crime to be brought to justice, especially when it results in pregnancy, regardless of who is the aggressor.

Although severely condemned in classic law texts, sexual crimes are considered as only weakly penalized by justice [22]. In judicial procedure, aggressor accountability (in sexual crimes) remains complex; being that most of the time, the expectation is to rely on evidence for the plea. Thus, in a Brazilian study with adolescents in court situations concerning sexual violence, condemnation of the aggressor was found to be significantly greater when the legal medical examination presented physical evidence of violence [23]. In this aspect, the forms of intimidation found in this study revealed that physical violence was significantly more frequent in sexual violence by strangers.

Taking advantage of the adolescent’s condition of vulnerability predominated in cases of incest. This reinforces the concept that there is a need within the family to grant invisibility to incest. To prevent identification of sexual violence and its consequent accountability, the abuser usually does not use physical violence, which significantly reduces the risk of producing physical evidence or victim resistance, a situation described by other authors [24]. Even in cases of incest in which vaginal or anal penetration occurs, physical injuries are found at low frequencies [25,26,27].

However, in cases of pregnancy by incest, our finding of a higher prevalence of adolescent intimidation (yet due to their condition of vulnerability) diverged from other studies which point to both intimidation and psychological threats as the most frequent forms of coercion [28,29]. In addition, the adolescent’s age, when less than 14 years, configured as a form of vulnerability which was significantly more frequent in cases of incest, emphasizing family dynamics and the victim’s physical inability to offer resistance as critically related phenomenon.

The adolescent’s consent or agreement in these situations was reflective of the emotional and psychological vulnerability in those younger than 14 years of age. Authors, such as Cohen and Gobetti (1998), consider consent questionable even for adult women, because the performance of incestuous desires reflects a lack of mental structure. The aggressor however is incapable of inhibiting his desires for concretely living experiences that should be suppressed [12].

The decision to abort an incestuous pregnancy is a complex issue for both the adolescent and her family; any decision remains subject to the strong influence of cultural, religious, and legal factors, which are widely different in each community or society. For example, Ramakuela et al. (2016) showed that in South Africa, incest was crucial to the decision to have an abortion among adolescents, since it is believed that keeping a pregnancy resulting from incest will curse the family and cause its rupture [30].

According to the World Health Organization, abortion consists in expulsion or embryonic fetal extraction occurring before the 20–22th week of gestation, when the estimated fetal weight is less than 500 g [31,32]. In gestations of less than 13 weeks, manual or electric intrauterine aspiration is one of the safest procedures to interrupt the pregnancy. From this gestational age onward, the method of choice is induction with medicines, particularly misoprostol in differing administration schemes which consider the gestational age [33].

The gestational age at admission for health care was significantly higher (concentrating above 12 weeks) among adolescents bringing allegations of pregnancy resulting from incest. This finding has important implications regarding both the possibility and method of performing an abortion. In allegations of pregnancy by incest, the percentage of adolescents with gestational ages of greater than or equal to 23 weeks was significantly higher. This prevented approval for the abortion request.

Approval of abortion was less frequent for adolescents having suffered sexual abuse from strangers. Pregnancy, as a result of a consensual relationship and sex with an intimate partner, whether antecedent or subsequent to the claimed sexual violence, was a verifiable condition; using incompatibilities between the date of sexual violence suffered reported, and (by obstetric ultrasonography), the estimated gestational age.

Gestational age should be considered a relevant factor in legal abortion assistance, (in cases of sexual violence against adolescents), since it occurs in a recognized period of intense physical and psychological changes, and increases the risk of negative consequences [34,35]. More comfortable methods of abortion, such as intrauterine aspiration, should be adopted whenever possible for adolescent girls; to avoid the trauma of the experience.

The results of this study reveal that during the 20 years of service provided by the Legal Abortion Referral Service, in the state of São Paulo, Brazil, an expressive number of alleged cases involving pregnancy by incest have been reported; this frequency has not been found in the literature in other regions of the country.

Further, the general expansion of the concept of health as based on social-ecological modeling of the personal and environmental factors affecting access to health services has emphasized factors and determinants involved in the processes of health and disease, and further, require guarantees of such access for female adolescent victims of rape. Incest delays arrival at health service facilities, and seems to be an interference factor for lawfully permitted abortion assistance. Incest corroborates to reduce opportunities for female adolescents victimized by rape to obtain full healthcare assistance.

This study is innovative for regarding the issue of pregnancy and its associated factors when resulting from incest. When considering the prevalence of such events and the extent of their consequences for the health of both families and society in general, this research underlines the need for further investigation by the scientific community.

## 5. Conclusions

Cases of pregnancy by incest presented indications that both proximity to the aggressor, and type of relationship can result in pregnancies at a very early (adolescent) age. In the incest group, the study also evidenced delayed procurement of public assistance, whether for police authorities or health services. In some cases, this prevented legal abortion. In the group (victims of strangers), the demand for services commonly appears sooner.

These findings interfere negatively in abortion assistance allowed by law, making it more frequent for incest adolescents to encounter, due to advanced gestational age, abortion as an unavailable option.

## Figures and Tables

**Table 1 medicina-55-00474-t001:** Socio-demographic and sexual violence characteristics of adolescents claiming pregnancy from sexual violence according to type of aggressor; Pérola Byington Hospital, 1994–2015, São Paulo, Brazil.

	Pregnancy						
Characteristic	By Stranger (*n* = 174)		By Incest (*n* = 137)		Total (*n* = 311)		
	n	%	n	%	n	%	*p **
**Religion**							
Yes	137	78.7	126	92.0	263	84.6	<0.001
No	37	21.3	11	8.0	48	15.4	
**Color or Ethnicity**							
White	81	46.5	66	48.2	147	47.3	0.776
Non-White	93	53.5	71	51.2	164	52.7	
**Approach**							
Public space	160	91.9	10	7.2	170	54.6	<0.001
Private space	14	8.0	127	92.7	141	45.3	
**Referral**							
Public security entity	114	65.5	109	48.8	223	71.7	0.006
Health service	40	22.9	25	38.4	65	20.9	
Spontaneous search	15	8.6	3	16.6	18	5.7	
Other	5	2.8	0	0.0	5	1.6	
**Intimidation**							
Violence	51	29.3	20	14.5	71	22.8	<0.001
Severe threat	24	13.7	21	15.3	45	14.4	
Violence and severe threat	61	35.0	30	21.8	91	29.2	
**Statutory rape**	39	21.8	66	48.1	105	33.4	
Statutory rape							
Age < 14 years	17	44.7	55	83.3	72	69.2	<0.001
Intellectual disability	7	18.4	8	12.1	15	14.4	
Drunkenness	6	15.7	0	0.0	6	5.7	
Substance acting on the CNS	9	23.6	3	4.5	12	11.5	
Total	39	100	66	100	105	100	
**Event Bulletin**							
Yes	147	84.5	124	90.5	271	87.1	0.115
No	27	15.5	13	9.5	40	12.9	
**IML Examination**							
Yes	140	80.4	116	84.7	256	82.3	0.334
No	34	19.9	21	15.3	55	17.7	

* Chi-Square; CNS: Central nervous system; IML: Instituto Médico Legal (Institute of Forensic Medicine).

**Table 2 medicina-55-00474-t002:** Median age, gestational age and number of sexual violence perpetrators for adolescents claiming pregnancy from sexual violence, as practiced by stranger or in a situation of incest; Pérola Byington Hospital, 1994–2015, São Paulo, Brazil.

	By Stranger	By Incest	
	Median (95% CI)		*p* *
Age	16 (16–16)	14 (14–14)	<0.001
Gestational age	14 (12–15)	18 (15–19)	<0.001
Number of perpetrators	1 (1–1)	1 (1–1)	0.002

* Mann-Whitney.

**Table 3 medicina-55-00474-t003:** Characteristics and outcomes of pregnancies among adolescents claiming pregnancy from sexual violence (according to the type of aggressor); and attended at Pérola Byington Hospital, 1994–2015, São Paulo, Brazil.

	Pregnancy						
	By Stranger (*n* = 174)		By Incest (*n* = 137)		Total (*n* = 311)		
	*n*	%	*n*	%	*n*	%	*p **
**Gestational age (weeks)**							
Up to 12	81	47.6	36	26.2	117	37.2	0.003
12–22	77	45.9	71	51.8	148	47.5	
≥23	16	4.2	30	21.8	46	14.7	
**Abortion**							
Yes	120	68.9	89	64.9	209	67.2	0.455
No	54	31.3	48	35.0	102	33.8	
**Method of abortion**							
Intra-uterine aspiration	61	50.8	35	39.3	96	45.9	0.198
Medical abortion	50	41.6	43	48.3	93	44.4	
Other method	9	7.5	11	12.3	20	9.5	
**Reason for not undergoing the abortion**							
Non-approval of the request	33	61.1	32	66.6	65	63.7	0.377
Waiver after approval	16	29.6	11	22.9	27	26.5	
Loss of follow-up	5	9.2	3	6.2	8	7.8	
Spontaneous abortion	0	0.0	2	4.1	2	2	
Total	54	100	48	100	103	100	
**Reason for non-approval of abortion**							
Gestational age ≥ 23 weeks	16	48.4	30	93.7	46	70.7	<0.001
Pregnancy not associated with violence	17	51.1	2	6.2	19	29.2	

* Chi-square.

**Table 4 medicina-55-00474-t004:** Characterization of Adolescent Pregnancy and Legal Abortion in Situations Involving Incest or Sexual Violence by an Unknown Aggressor, Poisson Regression, Brazil, 2019.

Variable	Prevalence	PR (95% CI)	*p* *
**Religion**			
No	22.9	Ref.	Ref.
Yes	47.9	2.09 (1.22–3.56)	0.007
**Approach**			
Public space	6.0	Ref.	Ref.
Private space	90.1	15.0 (8.21–27.5)	0.001
**Referral**			
Public security entity	48.9	Ref.	Ref.
Health service	38.5	0.78 (0.56–1.10)	0.162
Spontaneous search	16.7	0.34 (0.11–0.96)	0.043
**Intimidation**			
Violence	28.2	Ref.	Ref.
Severe threat	46.7	1.65 (1.01–2.69)	0.042
Violence and severe threat	33.0	1.17 (0.72–1.87)	0.515
Statutory rape	63.5		
**Statutory rape**			
Age <14 years	76.4	Ref.	Ref.
Intellectual disability	53.3	0.69 (0.42–1.14)	0.153
Drukenness	0.0		
Substance acting in the CNS	25.0	0.32 (0.12–0.88)	0.027
**Gestational age (weeks)**			
Up to 12	33.9	Ref.	Ref.
12–22	47.0	1.38(1.02–1.87)	0.033
≥23	68.0	2.0 (1.39–2.89)	<0.001
**Reason for non-approval**			
Gestational age ≥ 23 weeks	65.2	Ref.	Ref.
Pregnancy not resultingfrom violence	10.5	0.16 (0.04–0.61)	0.008

* Poisson Pregression; RP: Prevalence Ratio; CI: Confidence Interval; CNS: Central Nervous System; Ref: Referral.

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
