# Peer review of "Characterization of Adolescent Pregnancy and Legal Abortion in Situations Involving Incest or Sexual Violence by an Unknown Aggressor"

_medicina, 2019, doi:10.3390/medicina55080474_

Round 1

Reviewer 1 Report

In general, I have no comments for this clearly presented and interesting investigation. There were some phrasing that I found a bit odd, but this didn't detract from the overall quality of the manuscript (and the job of the reviewer isn't to focus on semantics, but to contribute to the scientific quality of the paper in my understanding). It is difficult to suggest any improvements as it is largely a descriptive study about incest vs rape. I did find that the authors went into too much detail on some aspects of the manuscript (for example, the discussion appears to be primarily a repeat of the results without adding a whole lot). If I was to make one recommendation, it would be to reduce the length of the discussion and focus more on a critical review of the results in comparison to previously published literature rather than just repeating the results.

Author Response

Dear Reviewer, Thanks for your feedback, we try to make improvements in writing.

Best regards

Reviewer 2 Report

The manuscript presents a complicated subject focusing on pregnancies in adolescents, which were the result of incest or sexual violence by stranger aggressors.

The study methodology includes medical records of the Perola Byington Hospital (San Paulo, Brazil) in the periods longer than 20 years.

Two groups of girls (i.e. adolescents with pregnancies from incest and resulting from rape by a stranger) were compared in various aspects, such as their characteristics (including religion, ethnicity), gestational age, percentage of abortion and its methods, reason for not undergoing the abortion.

The subject is very difficult in psychological, medical, legal, and social points of view.

However, there are some minor limitations:

Abstract:

- line 61 vs. line 64 – 134 or 137 pregnancies in the incest group (?)

- line 71 – please replace “pregnancy at a very early age” with “pregnancy at the age of 14”

Table 2:

- Does the group “by stranger” consist of girls aged 16 (with the median of 16 and 95% CI of 16-16 (??))? Were the girls at the age 14 (with the median of 14 and 95% CI of 14-14 (??)) included in the group “by incest”?

- If there is one perpetrator in the both groups, the statistical difference can not equal to 0.002 between the studied groups.

Conclusion:

- Please change the sentence in lines 353-354, because it is difficult to understand properly.

References:

References have not been described according to the instructions for authors.

Author Response

Dear Reviewer,

If there is one perpetrator in the both groups, the statistical
difference can not equal to 0.002 between the studied groups.”

All statistical tests were done more than once, and the data presented
correspond precisely to the result.

The other requests made about change of wording were made in lines
61-64, 71, 134-137. In addition to the adequacy of the conclusion, the
reference standards were also revised.